# Old and Novel Enteric Parvoviruses of Dogs

**DOI:** 10.3390/pathogens12050722

**Published:** 2023-05-16

**Authors:** Paolo Capozza, Alessio Buonavoglia, Annamaria Pratelli, Vito Martella, Nicola Decaro

**Affiliations:** 1Department of Veterinary Medicine, University of Bari Aldo Moro, 70010 Valenzano, Italy; paolo.capozza@uniba.it (P.C.); vito.martella@uniba.it (V.M.); nicola.decaro@uniba.it (N.D.); 2Department of Biomedical and Neuromotor Sciences, Dental School, Via Zamboni 33, 40126 Bologna, Italy; alessio.buonavoglia85@gmail.com

**Keywords:** dog, enteric viruses, emerging parvoviruses, protoparvoviruses, bocaparvoviruses, chaphamaparvoviruses

## Abstract

Parvovirus infections have been well known for around 100 years in domestic carnivores. However, the use of molecular assays and metagenomic approaches for virus discovery and characterization has led to the detection of novel parvovirus species and/or variants in dogs. Although some evidence suggests that these emerging canine parvoviruses may act as primary causative agents or as synergistic pathogens in the diseases of domestic carnivores, several aspects regarding epidemiology and virus–host interaction remain to be elucidated.

## 1. Introduction

Globally, gastroenteritis is considered a common clinical concern in domestic carnivores, chiefly in animals younger than one year of age and those residing in high-density animal environments, such as catteries and canine shelters. Viruses may be detected in 40–60% of diarrheal fecal samples from domestic dogs and cats, and they represent the main cause of gastroenteritis in these animal species [1,2,3,4,5]. Various viral agents, including astroviruses, adenoviruses, noroviruses, sapoviruses, rotaviruses, vesiviruses, kobuviruses and circoviruses have been identified in the feces of dogs and/or cats, associated with diarrhea either alone or in mixed infections, occasionally culminating in severe clinical signs. However, the most widely recognized primary causes of viral gastroenteritis in domestic carnivores are parvoviruses and enteric coronaviruses [3,4,5,6,7].

In recent years, using molecular assays and metagenomic approaches for virus discovery and characterization, various research groups have identified novel variants, lineages and species of parvoviruses in animals, resulting in a reclassification of the family *Parvoviridae* [7,8,9,10,11,12,13,14,15,16,17,18]. Parvoviruses (family *Parvoviridae*) are small (23 to 28 nm in diameter), nonenveloped viruses surrounded by an icosahedral capsid. They have a linear, single- stranded DNA genome, ranging from 4.5 to 5.5 kb (kilobases) in length, with complex hairpin structures at the 5′ and 3′ ends that are essential for viability [9]. The genome contains two open reading frames (ORFs). The first ORF encodes two non-structural (NS) proteins, NS1 and NS2, whilst the second ORF encodes two structural proteins, viral protein (VP) 1 and VP2 [8] (Figure 1).

NS1 is a pleiotropic nuclear phosphoprotein [19], essential for viral replication and responsible for inducing cell apoptosis [9]. VP2 is the major capsid protein and determines the viral tissue tropism and the host range [20,21,22,23]. Parvoviruses are considered members of the same species if their NS1 proteins contain more than 85% amino acid (aa) sequence identity, in accordance with the International Committee on Taxonomy of Viruses (ICTV) Classification standards. They may be categorized as the same taxon if their protein sequences cluster as a strong monophyletic lineage based on their entire NS1 protein sequence at the subfamily level and on their SF3 helicase domains at the family level. Moreover, NS1 proteins of members of the same genus should share at least 35–40% aa sequence identity, with a coverage of >80% between any two members. Along with the sequence identity-based criteria, common genus affiliation can also be justified based on a similar genome organization, i.e., the presence of auxiliary-protein-encoding genes, genome length and/or transcription strategy [10,11,24,25,26].

The *Parvoviridae* family is classified into three subfamilies, according to current ICTV guidelines, including *Parvovirinae* and *Densovirinae*, which infect vertebrates and arthropods, respectively, and the new subfamily *Hamaparvovirinae*, which infects both [10,11,12,25]. ICTV categorizes the subfamily *Parvovirinae* into eleven genera as follows: Amdoparvovirus, Artiparvovirus, Aveparvovirus, Bocaparvovirus, Copiparvovirus, Dependoparvovirus, Erythroparvovirus, Loriparvovirus, Protoparvovirus, Sandeparvovirus and Tetraparvovirus (Table 1) [10,11,12,25,27]. Unknown parvoviruses have been found in a variety of domestic and wild species including dogs, sea otters, seals, bats, camels, horses and pigs [28,29,30,31,32,33,34].

Little is known about the epidemiology and genetic variability of these novel parvoviruses. It is unclear if they can cause enteric infections in dogs and to what extent they can impact canine health.

The purpose of this study is to offer an update on canine parvoviruses that have recently been found in relation with gastrointestinal signs.

## 2. Protoparvoviruses

The species *Protoparvovirus Carnivoran 1* (Table 1), within the genus *Protoparvovirus*, includes genetically and antigenically related viruses such as canine parvovirus type 2 (CPV-2), feline panleukopenia virus (FPV) and parvoviruses of wild animals, all of which cause severe diseases, especially in young animals [7,10,11,12,27,42,43]. The phylogenetic relationships between CPV-2 isolates from dogs and the viruses isolated from cats (FPV), minks (mink enteritis virus, MEV), raccoons (raccoon parvovirus, RPV), raccoon dogs (raccoon dog parvovirus, RDPV) and blue foxes (blue fox parvovirus, BFPV) showed that all these viruses belong to the species *Protoparvovirus Carnivoran 1* and derive from a single common ancestor [44,45]. Due to a >99% nucleotide (nt) genomic identity with FPV, it was suggested that CPV-2 might be originated from a close relative of FPV through the accumulation of punctate mutations [44]. Key amino acid (aa) residues (80R, 93N, 103A, 323N, 564S and 568G) in VP2 protein distinguish CPV-2 from FPV [42,46,47,48,49,50]. CPV-2 likely arose by acquiring mutations that allowed binding to the canine transferrin receptor (TfR) type-1 [45,51,52]. Several studies demonstrated that TfR plays a key role in the susceptibility of cells to infection by these viruses [20,53,54]. The evolutionary pathways and inter-species jump of protoparvoviruses in carnivores have stimulated a high scientific interest worldwide.

### 2.1. CPV-2 Variants

CPV arose as a pathogen of canines in the late 1970s when it was responsible for a global panzootic in dogs of all ages, which, at that time, were naïve to the infection [42]. CPV is considered remarkably contagious and causes significant morbidity with increased incidence in shelters, pet stores and breeding kennels. A rapid clinical course defines the disease with mortality often occurring 2–3 days after the onset of signs in nonprotected animals [55,56]. The infection is generally acquired by the fecal–oral route through contact with fecal material from infected animals or contaminated surfaces [5]. The virus mainly infects mitotically active tissues, such as the lymphoid tissues, intestinal epithelium and bone marrow, and the heart in neonatal puppies. Following an incubation period of 3–7 days, an enteric form is observed that is characterized by vomiting, hemorrhagic diarrhea, depression, loss of appetite, fever and dehydration in younger animals [23,55,56,57]. The disease can affect dogs at any age, but severe infection is most common in puppies between 6 weeks and 6 months of age [58], with all breeds being susceptible to the disease [58,59].

The original strain was named CPV-2 to distinguish it from the genetically and antigenically unrelated canine parvovirus type 1 (CPV-1, also known as canine minute virus, CnMV or MCV), which has been recently reclassified as *Bocaparvovirus Carnivoran 1* (genus *Bocaparvovirus*) (Table 1) and is associated with neonatal mortality [42,60]. A few years after its appearance, CPV-2 generated a first antigenic variant, CPV-2a, which differs from the original type-2 in 5–6 aa sites of the capsid (VP2) protein. A second antigenic variant, CPV-2b, exhibited a further mutation in the VP2 (from asparagine to aspartic acid (N to D) at aa residue 426) [61,62,63]. In 2000, a third antigenic variant, CPV-2c, was identified, which exhibited the aa variation asparagine/aspartic acid (N/D) to glutamic acid (E) at residue 426 of the VP2 protein [64]. A single aa variation among CPV-2a, -2b and -2c confers distinct antigenic properties, as indicated by the diverse reactivity to particular monoclonal antibodies [65]. However, the variants lack distinct monophyletic segregation due to additional point mutations in various regions of their genome and encoded proteins [66].

The three variants are disseminated globally and are enduring constant evolution. After 1980, CPV-2 variants completely supplanted the original CPV-2 or CPV-2-like virus in the field, although the original CPV-2 type is still present in some vaccine formulations [49,65,67,68,69,70,71,72,73,74,75,76,77,78,79,80,81]. In most countries, only two CPV types are included in the modified live virus (MLV) vaccine formulations, the original CPV-2 strain and its variant CPV-2b, which are able to induce immunity against all three CPV-2 variants. Both MLV and CPV vaccine strains can cause viremia and can replicate in the intestinal mucosa, albeit at lower titers than the field strains, being excreted in the feces of vaccinated animals for at least 3–4 weeks post-vaccination. On the other hand, a few inactivated CPV vaccines are available on the market since these formulations have low immunogenicity, and their use is suggested only in wildlife animals and pregnant female dogs [66].

A recent study revealed that the CPV-2c proportion has been increasing gradually, substituting CPV-2a as the new dominant variant since 2020 [82]. In addition, CPV-2b maintained a low epidemic relevance with a peak of circulation in 2003. However, the dynamic changes in CPV-2 variants may differ geographically. In Asia, CPV-2a has long been the dominant strain, but it was replaced by CPV-2c in 2020. In Europe, CPV-2a/2b/2c are co-epidemic, and the circulation rate of CPV-2-like viruses is low. In 2004, CPV-2a superseded CPV-2c as the dominant variant in South America. In Oceania, there was a progressive change from CPV-2a to a co-endemic circulation of CPV-2a and CPV-2b. Before 2014, CPV-2 variants in North America co-circulated without any apparent epidemiological trend, but CPV-2c steadily became predominant from 2014 onwards. Likewise, based on the temporal dynamic patterns, after 2017, CPV-2c appeared to replace CPV-2a in Asia, South America and Africa, but not in Europe and Oceania [82,83,84].

CPV-2 VP2 is the major component of the viral envelope. Several VP2 aa residues are related to antigenicity and host range [85], and antigenic drift may account for vaccine failure [86,87]. Previously, it was found that three aa mutants in VP2 (F267Y, Y324I and T440A) are likely involved in vaccination failures [88]. Monitoring of the various patterns of these three aa residues has shown that strains with the mutations 267Y and 324I have become predominant. In contrast, the frequency of 440A did not exceed that of the original residue 440T, which peaked in 2014 and then steadily fell afterward [82].

Recent analyses of aa mutations in the VP2 gene of the Asian CPV-2c strains revealed two frequent mutations, A5G and Q370R [82,89]. The A5G mutation was initially discovered in China in 2015 [90]. The Q370R mutation was first identified in CPV-2a strains detected from giant pandas in Sichuan, China, and subsequently became the predominant mutation in CPV-2c strains [91]. It is still unclear if the vaccines used for CPV prophylaxis provide adequate protection against the 5G and 370R mutant CPVs [82].

In summary, it is pivotal to monitor the evolution of VP2 to promptly recognize the appearance of novel variants of CPV-2. Some immunized dogs may develop the disease and it is unclear if this is accounted for by CPV-2 variants with mutations in key residues [51,66,92,93]. Overall, understanding the molecular and biological properties and epidemiological trends of CPV-2 could help to prevent and control parvovirus disease.

Compared with the original CPV-2 strain, the variants have likely reached a greater fitness in the canine hosts. Additionally, they have re-gained the feline host range, and in cats, they can determine subclinical infection or diseases indistinguishable from FPV-induced feline panleukopenia [24,43].

### 2.2. FPV and FPV-Like

The transmission of CPV-2 to cats and of FPV to dogs has stimulated the research. Experiments in vivo and in vitro were conducted to decipher the mechanisms controlling the host range of these viruses [94]. FPV seems to replicate effectively only in feline cells, whilst CPV-2 can be cultured in both canine and feline cells [22,36,57,94,95,96].

The mechanisms ruling host ranges in vivo of FPV and CPV-2 have been recently disentangled. FPV replicates in feline tissues, including the thymus, spleen, lymph nodes and intestinal epithelial cells, and virus load in the feces is very high. In dogs, FPV replication is restricted to the thymus and bone marrow [36].

Under natural conditions, all CPV-2 variants were identified, although sporadically, from cats with feline panleukopenia with several independent reports from different countries [43,51,97,98,99,100,101]. On the contrary, reports on FPV in dogs are less frequent. In 1993, an isolate was obtained from a dog with acute gastro-enteritis, but the virus had features more similar to FPV [100]. More recently, the transmission of FPV to dogs with enteric clinical signs was reported in Pakistan, Thailand, Vietnam, China and Italy [79,94,95,102,103,104]. These viruses were characterized as FPV after either the partial or complete sequence analysis of the capsid gene. The aa mutation from K to N at VP2 residue 93, involved in host range control, was observed in an FPV-like strain detected from a dog in Thailand [103] and the I101T mutation was identified in FPV-like strains detected from dogs in Vietnam [94], China [95] and Italy [104]. The pathogenetic role, if any, of FPV in dogs remains unclear. Although the circulation of FPV or FPV-like viruses has been repeatedly documented in dogs, FPV seems to infect dogs only occasionally. Residues in the apical domain of TfR seem critical for controlling parvovirus binding [20,53,105,106]. Sequencing of the canine TfR from FPV-infected dogs could eventually help to gain an understanding of the ability of FPV to infect some dogs [50]. Likewise, genome sequencing could be useful to monitor the evolution of *Protoparvovirus Carnivoran 1* variants in domestic and wild carnivores [104].

### 2.3. Canine Bufavirus

In 2018, a new protoparvovirus strain was detected in a litter of five-month-old puppies involved in an outbreak of a respiratory disease [32]. This virus was provisionally named canine bufavirus (CBuV) and displayed a low aa 19.3–51.4% identity in the NS1 to members of the species *Protoparvovirus Carnivoran 1*, while the closest relatives to CBuV (47.2–51.4% aa identity in NS1) were protoparvoviruses identified in human and non-human primates, commonly termed as bufaviruses (BuVs) [107,108,109]. Following the ICTV classification criteria, the canine BuV was classified as a putative novel species, *Protoparvovirus Carnivoran 3* (Table 1)*,* within the genus *Protoparvovirus* [32,110].

In humans, and more recently, in wild animals (wolves and foxes), BuVs have been detected mostly in the gastrointestinal tract [111,112]. Studies in dogs [32,113], non-human primates [109], shrews [114] and sea otters [33] also suggest possible extraintestinal and/or systemic infections. In 2019, in China, canine BuVs were detected in sera from dogs with respiratory signs [113].

The circulation of CBuV in the canine population was described in Italy [32,115], China [113,116] and India [117], but its genetic and pathobiological features are still unclear [118].

In Italy, the detection rate of this virus was 7.7% (16/207), with a higher frequency (8.8%) in diarrheic dogs, but CBuV infection was non-statistically correlated with gastrointestinal disease [115]. In China, CBuV was identified in Shanghai, Guangxi province and Henan province, with prevalence rates as high as 42.15% (51/121), 2.5% (5/200) and 1.74% (2/115), respectively [113,116,119]. Another study in the Chinese province of Anhui revealed a CBuV prevalence of 2.5% (3/120) [118]. In a more recent investigation, CBuV was detected with a proportion of 4.3% (8/186) from both diarrheic puppies (<1 year old) and adult dogs (>1 year old) [117]. On genome sequencing, the strain 407/PVNRTVU/2020 showed a 93.4–98.8% nucleotide (nt) identity to other CBuV sequences, and it was closely related to other CBuV strains detected in China [117].

The primary clinical sign caused by most species of *Protoparvovirus* in carnivores is diarrhea [5]. Some studies revealed a positive correlation between CBuV infection and diarrhea, and CBuV DNA was also detected in the serum samples of dogs with gastroenteritis [116]. The genome sequencing of CBuV strains demonstrated genetic heterogeneity and suggested that recombination may be important factors in the virus’s evolution [115].

In most cases, the CBuV genome was detected in dogs in co-infection with other viral pathogens including CPV-2 [115,118], canine coronavirus (CCoV), canine kobuvirus (CKoV) [115], canine adenoviruses type 1 (CAdV-1) and type 2 (CAdV-2) [117], suggesting that CBuV could be considered as a common component of canine fecal virome. Although the pathobiology of CBuV in dogs remains unclear, a possible role of this virus in the etiology of canine enteritis can be hypothesized. Synergistic effects after co-infection with other enteric viruses could increase the severity of clinical signs.

As observed for CPV-2 and FPV, there is evidence that CBuV can also infect the feline host [110,120].

The ability of viruses in the genus *Protoparvirus* to determine severe clinical signs in dogs, as well as their multi-host nature, must be considered in the implementation of prophylaxis plans, to limit the spread of these viruses not only among individuals of the same species, but also between individuals of different species sharing the same environment.

## 3. Bocaparvoviruses

The genus *Bocaparvovirus* (BoV), in the subfamily *Parvovirinae*, includes viruses detected or associated with diseases in various animals, including pigs [121,122], cattle [123], California sea lions [124], bats [125], rabbits [126], rodents [127], pine martens [128], minks [129], dogs [37,38,39], cats [13,14,38,130], gorillas [131,132] and in humans (HBoVs) [133,134], suggesting a potentially wide host range of these viruses.

Viruses in this genus are monophyletic and have >30% NS1 aa identity. However, <30% identity values are allowed between certain viruses to accommodate disparities between current and previous analytical methods. BoVs are unique among parvoviruses since they possess an additional ORF (ORF3), located between the non-structural (ORF1) and structural (ORF2) coding regions of their genome (5.5 kb ssDNA) (Figure 2).

ORF3 encodes the NP1, a highly phosphorylated protein different from proteins of other parvoviruses and involved in RNA processing. NP1 regulates VP-encoding RNAs’ splicing and read-through of the proximal polyadenylation [27,123,135,136,137,138,139]. The great majority of these viruses were identified using metaviromic strategies and they have not been adapted to culture systems [10,11,27].

The genus was originally named after two prototype members, bovine parvovirus (BPV) and CnMV (formerly known as CPV-1) [140]. Based on the ICTV classification criteria, BoVs are classified into thirty-two species, of which at least six species were detected in domestic carnivores and classified as *Bocaparvovirus Carnivoran* (CBoV) 1-5 and 7, while another species (CBoV-6) was found in minks (Table 1) [11,12,27,129]. Currently, BoVs identified in domestic dogs are classified within the species CBoVs-1, 2 and 7 [13,14,38] (Table 1), and they are associated with different clinical manifestations.

As previously mentioned, CnMV is an autonomous parvovirus of dogs that is genetically and antigenically unrelated to CPV-2 [42]. CnMV is currently classified as the species *Bocaparvovirus Carnivoran-1* [141]. CnMV was first isolated in 1967 from the feces of a clinically healthy military dog [35,142], and it seems common in domestic dogs globally. The clinical significance and pathogenicity of CnMV are debated. CnMV causes mild infections in puppies, and it seems weakly pathogenic in adults [140]. Pneumonitis, hepatitis, myocarditis and lymphadenitis were described in dogs infected with CnMV [60,143,144]. CnMV may cross the placental barrier, resulting in early fetal death, birth defects and neonatal mortality [60,143,145].

*Bocaparvovirus Carnivoran-2* (CBoV-2) was first identified in 2012 in dogs with canine respiratory disease [39]. CBoV-2 share less than 63%, 62% and 64% aa identity with CnMV in the NS, NP and VP genes, respectively [39]. CBoV-2 infection was detected in pups with severe enteritis characterized histologically by atrophied and fused villi, intense crypt regeneration and severe bone marrow and lymphoid atrophy [28]. CBoV-2 infection has also been associated with interstitial pneumonia [146].

Variants of CBoV-2 have been detected in fecal, nasal, urine and blood samples collected from dogs in Hong Kong [38], thus suggesting possible extraintestinal and/or systemic infections. A more recent investigation identified a novel strain of CBoV-2 in a litter of puppies in Thailand that died from acute dyspnea and hemoptysis [147]. This strain was more closely related to CBoV-2 strains identified previously in South Korea [146] and Hong Kong [28].

Using deep sequencing, a third type of canine bocavirus, CBoV-3 (currently proposed as *Bocaparvovirus Carnivoran*-7) (Table 1) [27], was identified in 2013 in the liver of a dog with hemorrhagic gastroenteritis, necrotizing vasculitis, granulomatous lymphadenitis and anuric renal failure, and coinfected with a canine circovirus [40]. CBoV-3 is classified as an additional novel species, since in the NS1, NP1 and VP1 regions, it shares only 49–51%, 52–57% and 56–57% aa identity, respectively, with CBoV-1 and CBoV-2 [40]. On the phylogenetic analysis of the complete VP1, CBoV-3 is phylogenetically distinct from other canine bocaparvoviruses [40]. Episomal forms were detected via PCR, indicating that replication may occur in hepatocytes or other liver cell types. Since circovirus infection can lead to lymphocyte depletion and immunosuppression in the host [148], the pathogenic role of CBoV-3 in the original study could not be assessed clearly [40].

Including CBoVs in the diagnostic algorithm of canine enteritis, using specific molecular tools could help to better understand the enteropathogenic role of these viruses and to assess whether some CBoV species/strains possess peculiar phenotype changes.

Finally, it is noteworthy to mention that several bocaparvoviruses were detected in cats and other wildlife carnivores [13,14,24,38,129,130,149].

## 4. Chaphamaparvoviruses

The genus *Chaphamaparvovirus* (ChPV) is included in the subfamily *Hamaparvovirinae* (Table 1). This genus comprises viruses that are genetically more related to parvoviruses of invertebrates. As long as additional ChPV sequences are generated, this classification could be updated [11,12,25,27].

ChPV was first identified in an oropharyngeal swab sample collected from a fruit bat (*Eidolon helvum*) in Ghana (Africa) [150]. Subsequently, ChPV-like viruses were described in other animal species [25], including dogs and cats [15,16,18,41,151]. The first report of ChPV in pets dates back to 2017, in USA, in a metaviromic study carried out on the feces of dogs with hemorrhagic enteritis [41]. Viruses genetically related to the American canine ChPV prototypes (provisionally termed as cachavirus) were subsequently detected in the stools of pets in China and Italy [15,18,151]. Based on the ICTV classification outlines, all strains of canine origin segregate into the new species *Carnivore chaphamaparvovirus 1* [11]

A possible association of ChPV with enteric disease in dogs was first hypothesized in a 2019 study [41]. However, this possible association has not been demonstrated in other studies [18,151]. In a recent investigation in Thailand, a correlation was observed with the presence of ChPV DNA in the stools of young dogs with mild enteritis but not in archival samples of deceased animals with diarrhea [152].

Although the presence of ChPV DNA in cases of canine enteric disease has been documented repeatedly [18,41,151,152], information about CaChPV-1 tropism and viral distribution in the intestine or in other organs is limited. 

Epidemiological and clinical information are required to understand if ChPVs have a role as enteric pathogens in dogs [18,41,151,152]. Indeed, ChPVs’ DNA has been often detected in dogs in co-infection with other viral pathogens such as CBuV, CAdV [153], CPV-2, CCoV [18,151] and canine distemper virus (CDV) [18].

## 5. Conclusions

In the last twenty years, the exploration of canine viromes using sequence-independent protocols and consensus (pan-viral) PCR strategies has identified several novel parvovirus species and variants in dogs with enteric and/or respiratory diseases. Whether these novel canine parvoviruses may act as primary causative pathogens or synergistic agents remain to be elucidated.

A trend in the diagnostics of human infectious disease is the adoption of syndromic testing panels covering a wide spectrum of common and uncommon pathogens based on advanced microbiology technologies such as multiplex molecular assays (i.e., syndromic diagnostic tests). Including these novel canine parvoviruses in the diagnostic algorithms of canine diseases, combined with larger epidemiological studies with a multidisciplinary approach and/or with experimental infections, could help to clarify their epidemiology and their eventual association, if any, with canine diseases. Expanding our knowledge on the enteric viromes of animals at the animal–human interface is, by the way, necessary to assess more properly eventual zoonotic risks and fulfill the recommendations of the One Health paradigm.

Interestingly, the multi-species circulation of some of these novel parvoviruses could represent a challenge when devising measures of prophylaxis in animals of different species living/housed in the same household, shelters and clinics.

The use of vaccines, when available, could prevent the spread of many of these emerging parvoviruses, although this strategy should be complemented with detailed disinfection plans and the physical separation of animals, chiefly in the case of suspected parvovirus circulation in multi-animal and multi-species environments.

## Figures and Tables

**Figure 1 pathogens-12-00722-f001:**
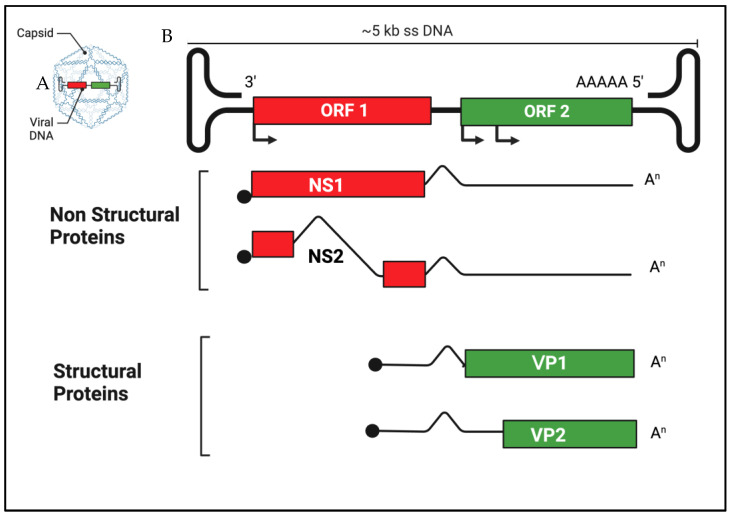
Schematic representation of canine parvovirus prototype (CPV-2, genus Protoparvovirus) (**A**). ORFs organization of genome and its transcriptional strategy (**B**). The genome presents terminal palindromic sequences that form complex hairpin structures at the 5′ and 3′ ends. These structures serve as the origin of DNA replication and help encapsidation (packaging) of viral DNA. The 5′ ends of RNA transcripts are capped (black circles), and the 3′ ends are polyadenylated (An). VP1 and VP2 are encoded in the same mRNA.

**Figure 2 pathogens-12-00722-f002:**
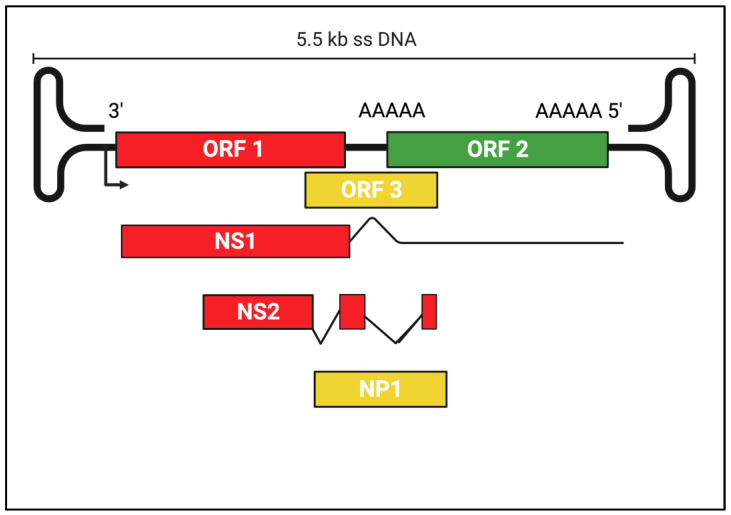
Canine Bocaparvovirus: ORF genetic organization and replication strategy. The genome is depicted as a single line terminating in enclosed hairpin structures. Segments encoding the non-structural proteins (NS1, NS2) are shaded in red, the capsid protein (ORF2) is shaded in green, and the NP1 (ORF3) is in yellow. A solid arrow denotes the RNA polymerase-II promoter; polyadenylation sites are indicated by the letters AAAAA.

**Table 1 pathogens-12-00722-t001:** Parvoviruses detected in dogs and their up-to-date classification.

Subfamily	Genus	Species	Common Names Used in the Literature	Country of First Identification	Year	Detection Source	Reference
*Parvovirinae*	*Protoparvovirus*	*Protoparvovirus* *Carnivoran 1*	Canine Parvovirus type 2 (CPV-2 and variant a, b, c)	USA	1977	Stools	[35]
			Feline Panleukopenia Virus (FPV-like)	Germany	1992	Thymus, bone marrow	[36]
		*Protoparvovirus* *Carnivoran 3*	Canine Bufavirus (CBuV)	Italy	2018	Stools, respiratory samples	[32]
	*Bocaparvirus*	*Bocaparvovirus* *Carnivoran 1*	Canine Parvovirus type 1 (CPV-1), Canine Minute Virus (CnMV/MCV), Canine Bocavirus 1 (CBoV-1)	USA	1967	Stools	[37]
		*Bocaparvovirus* *Carnivoran 2*	Canine Bocavirus 2 (CBoV-2)	USA	2012	Stools, respiratory samples	[38,39]
		*Bocaparvovirus* *Carnivoran 7*	Canine Bocavirus 3 (CBoV-3)	USA	2013	Liver	[40]
*Hamaparvovirinae*	*Chaphamaparvovirus*	*Carnivore* *chaphamaparvovirus 1*	Canine chaphamaparvovirus-1 (CChPV-1)	USA	2017	Stools	[41]

## Data Availability

Not applicable.

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
