# Peer review of "Old and Novel Enteric Parvoviruses of Dogs"

_pathogens, 2023, doi:10.3390/pathogens12050722_

Round 1

Reviewer 1 Report

The authors provide a brief description of the known enteric parvoviruses of dogs. The manuscript is informative and clearly written. The complete lack of figures (such as a philogenetic tree or a graphical comparison of certain structural features between different parvoviruses) makes it more difficult to read and less attractive to virologists who are not specialists on parvoviruses or virus taxonomy.

There are a few typos (e.g., lines 66-67 where the last sentence of section 1. (Introduction) is in bold face and merged wit the heading for Section 2. (Protoparvovirus).

Author Response

Comments from Reviewer # 1

R 1.1: The authors provide a brief description of the known enteric parvoviruses of dogs. The manuscript is informative and clearly written. The complete lack of figures (such as a philogenetic tree or a graphical comparison of certain structural features between different parvoviruses) makes it more difficult to read and less attractive to virologists who are not specialists on parvoviruses or virus taxonomy.

Reply to R 1.1: We agree with the referee and have incorporated your suggestion in the revised manuscript adding Figure 1 and Figure 2 in which we reported a graphical schematization of viral genome and the replication strategies.

R 1.2: There are a few typos (e.g., lines 66-67 where the last sentence of section 1. (Introduction) is in bold face and merged with the heading for Section 2. (Protoparvovirus).

Reply to R 1.1: We agree with the referee and have incorporated your suggestion throughout the manuscript. We have revised the formatting issue lines 66/67.

Reviewer 2 Report

This is a very comprehensive and up-to-date review of viruses from the Parvoviridae family that have been detected in dogs. The authors provide detailed information on both the molecular biology and epidemiology of these viruses. The limitations of current studies are carefully discussed, and areas where further research is required are neatly highlighted. I have a few minor comments as listed below, but overall I think this will be a valuable review for readers interested in this field.

Table 1: This is a very useful table, matching the ICTV nomenclature with common names used in the literature. Minor changes recommended:

-       Correct ‘Texas’ in the ‘country of first identification’ column to USA.

-       Suggest editing title of this table: Canine parvovirus type 2 would not be considered ‘emerging’.

Section 2.1 CPV-2 variants: It would be interesting to include a short summary of which variants are currently in mainstream vaccines, and how protective these are for heterologous variants. This is briefly touched upon in line 116, and again in line 132,  but more information would be useful for more clinically focused readers.

What are the most likely reasons for geographical variation in variants? Is there any evidence of vaccine pressure changing which variants are dominant?

Abstract: correct ‘dog’ to ‘dogs’ line 11

Please check formatting issue lines 66/67

Author Response

Comments from Reviewer # 2

R 2.1: This is a very comprehensive and up-to-date review of viruses from the Parvoviridae family that have been detected in dogs. The authors provide detailed information on both the molecular biology and epidemiology of these viruses. The limitations of current studies are carefully discussed, and areas where further research is required are neatly highlighted. I have a few minor comments as listed below, but overall, I think this will be a valuable review for readers interested in this field.

Table 1: This is a very useful table, matching the ICTV nomenclature with common names used in the literature. Minor changes recommended:

-       Correct ‘Texas’ in the ‘country of first identification’ column to USA.

-  Suggest editing title of this table: Canine parvovirus type 2 would not be considered ‘emerging’.

 Reply to 2.1: We agree with the referee and have incorporated your suggestion throughout the revised manuscript. We correct “Texas” with “USA”, and we changed the title of the table 1.

R 2.3: Section 2.1 CPV-2 variants: It would be interesting to include a short summary of which variants are currently in mainstream vaccines, and how protective these are for heterologous variants. This is briefly touched upon in line 116, and again in line 132, but more information would be useful for more clinically focused readers.

Reply to 2.3: We agree with the referee and have incorporated your suggestion throughout the revised manuscript, adding just some spot information regarding the CPV vaccine types and the role of the CPV-2 variants in immunization failures. However, we remind you that the purpose of this review is not to evaluate and provide information on the various vaccines against CPV available on the market and the various vaccination protocols made available. Therefore lines 139-147 are given general information on the matter.

R 2.4: What are the most likely reasons for geographical variation in variants? Is there any evidence of vaccine pressure changing which variants are dominant?

 Reply to 2.4: The most likely reasons for geographical variation in variants are probably the natural interaction between the host and the virus, with natural variation drift events, and in particular, the possibility for naïve subjects (domestic or wild carnivores) to become infected with other variants. As reported by Altman et al., 2017, there is no strict correlation between immunization failures and the antigenic CPV type contained in the vaccines, whereas the leading risk factor identified was early termination of the primary vaccination course schedules.

R 2.5: Abstract: correct ‘dog’ to ‘dogs’ line 11

 Reply to 2.5: We agree with the referee and have incorporated your suggestion throughout the manuscript.

R 2.6: Please check formatting issue lines 66/67

Reply to 2.6: We agree with the referee and have incorporated your suggestion throughout the manuscript. We have revised the formatting issue in lines 66/67.